# Associations of Serum Folate and Holotranscobalamin with Cardiometabolic Risk Factors in Rural and Urban Cameroon

**DOI:** 10.3390/nu14010178

**Published:** 2021-12-30

**Authors:** Camille M. Mba, Albert Koulman, Nita G. Forouhi, Fumiaki Imamura, Felix Assah, Jean Claude Mbanya, Nick J. Wareham

**Affiliations:** 1MRC Epidemiology Unit, School of Clinical Medicine, University of Cambridge, Cambridge CB2 0QQ, UK; ak675@cam.ac.uk (A.K.); nita.forouhi@mrc-epid.cam.ac.uk (N.G.F.); fumiaki.imamura@mrc-epid.cam.ac.uk (F.I.); nick.wareham@mrc-epid.cam.ac.uk (N.J.W.); 2Department of Public Health, Faculty of Medicine and Biomedical Sciences, University of Yaoundé 1, Yaoundé P.O. Box 1364, Cameroon; kembeassah@yahoo.com; 3Nutritional Biomarker Laboratory, National Institute for Health Research Biomedical Research Centre, School of Clinical Medicine, University of Cambridge, Cambridge CB2 0QQ, UK; 4Department of Internal Medicine and Specialties, Faculty of Medicine and Biomedical Sciences, University of Yaoundé I, Yaoundé P.O. Box 1364, Cameroon; jcmbanya@yahoo.co.uk

**Keywords:** serum folate, holotranscobalamin, cardiometabolic risk factors

## Abstract

A low intake of fruit and vegetables and a high intake of meat are associated with higher cardiometabolic disease risk; however much prior research has relied on subjective methods for dietary assessment and focused on Western populations. We aimed to investigate the association of blood folate as an objective marker of fruit and vegetable intake and holotranscobalamin (holoTC) as a marker of animal-sourced food intake with cardiometabolic risk factors. We conducted a population-based cross-sectional study on 578 adults (mean ± SD age = 38.2 ± 8.6 years; 64% women). The primary outcome was a continuous metabolic syndrome score. The median serum folate was 12.9 (IQR: 8.6–20.5) nmol/L and the mean holoTC was 75 (SD: 34.3) pmol/L. Rural residents demonstrated higher serum folate concentrations (15.9 (9.8–25.9) nmol/L) than urban residents (11.3 (7.9–15.8) nmol/L), but lower holoTC concentrations (rural: 69.8 (32.9) pmol/L; urban: 79.8 (34.9)) pmol/L, *p* < 0.001 for both comparisons. There was an inverse association between serum folate and metabolic syndrome score by −0.20 in the z-score (95% CI, −0.38 to −0.02) per 10.8 (1 SD) of folate) in a model adjusted for socio-demographic factors, smoking status, alcohol intake, BMI, and physical activity. HoloTC was positively associated with the metabolic syndrome score in unadjusted analysis (0.33 (95% CI, 0.10 to 0.56)) but became non-significant (0.17 (−0.05 to 0.39)) after adjusting for socio-demographic and behavioural characteristics. In conclusion, serum folate and holoTC were associated with the metabolic syndrome score in opposite directions. The positive association between serum holoTC and the metabolic syndrome score was partly dependent on sociodemographic characteristics. These findings suggest that, based on these biomarkers reflecting dietary intakes, public health approaches promoting a higher intake of fruit and vegetables may lower cardiometabolic risk factors in this population.

## 1. Introduction

Cardiometabolic diseases are a cluster of disorders linked to insulin resistance, hypertension, dyslipidaemia, and abdominal obesity [1]. These diseases are amongst the leading causes of premature mortality and disability worldwide, with most of the deaths occurring in low- and middle-income countries. Over the last decades, the prevalence of cardiometabolic diseases has increased globally, but at a faster rate in low- and middle-income countries [2]. Identifying the modifiable determinants of cardiometabolic disease is a public health priority.

According to the 2017 global burden of disease study, low intake of fruit and vegetables was amongst the leading dietary risk factors for non-communicable disease deaths in sub-Saharan Africa (SSA) [3]. In addition, a meta-analysis of nine prospective studies showed that higher meat consumption (particularly red and processed meat) was linked to a higher risk of the metabolic syndrome [4]. A few cross-sectional studies in SSA have reported an inverse association between fruit and vegetable intake and hypertension [5,6] and no association between meat intake and hypertension [7]. The results of a meta-analysis of 47 studies from 22 SSA countries suggest fruit and vegetable intake is low in SSA, with 79.1% of adults reporting fruit and vegetable intakes below the World Health Organisation (WHO) recommended minimum daily consumption of 400 g, while meat consumption is high (51% of adults consuming >70 g daily) [8]. However, these studies have relied on self-reported dietary assessment methods, which are subject to recall bias and measurement error.

Circulating plasma vitamin C and carotenoids have been widely investigated as objective biomarkers of fruit and vegetable intake [9,10] and applied to test diet–disease association [11]. However, fruit and vegetables are also rich sources of folate. Plasma folate has been shown to be correlated with self-reported intake of fruit and vegetables in observational studies [12]. In short-term (4 days to 3 weeks) intervention studies of fruit and vegetable intake compared to control conditions, plasma folate increased [13,14,15]. Natural vitamin B12 is found almost exclusively in animal products. Intake of animal products has been shown to be positively correlated with total vitamin B12 and holotranscobalamin (holoTC) in observational and intervention studies [16,17]. HoloTC is vitamin B12 bound to transcobalamin II, and is the metabolically active fraction of vitamin B12 (fraction that can readily enter the cells). Some studies suggest holoTC is a better indicator of vitamin B12 status than total vitamin B12 [18]. Taken together, this evidence suggests that blood folate may be a practical and valid biomarker of intake of fruit and vegetable and that blood vitamin B12 can be used as a marker of animal-sourced foods especially in countries where food fortification with folate or vitamin B12 is not mandatory.

Although blood folate and vitamin B12 have been proposed as objective indicators of intake of fruit and vegetable and animal-sourced foods respectively, we did not find any previous studies of these biomarkers in SSA. Our study aimed to examine how serum folate and holoTC could be related to the clustering of cardiometabolic risk factors in adults in rural and urban Cameroon. Given that food fortification with folate or vitamin B12 was not mandatory in Cameroon at the time of data collection, this study offers the possibility to examine the associations between dietary patterns for which these B-vitamins are objective indicators and cardiometabolic risk factors.

## 2. Methods

The methods used for the recruitment of the participants in this population-based study have been described elsewhere [19]. Briefly, participants aged between 25–55 years without a history of diabetes or cardiovascular diseases were recruited in Cameroon between January 2005 and December 2006 from two urban sites (Yaoundé, the capital city of Cameroon in the Centre region, and Bamenda, the capital city of the North West region) and two rural sites (Mbankomo in the Centre region and Bafut in the North West region). This cross-sectional study initially recruited 651 participants. For the current analysis, we excluded participants who did not have measures of serum folate or holoTC. In total, data on 578 and 547 participants were available for the current analysis of serum folate and holoTC respectively. The national ethics committee approved the study and all the participants provided written informed consent. 

### 2.1. Serum Folate and Holotranscobalamin Measurement

A blood sample from each participant was collected at a fasting state. Following centrifugation at ~1400*g*, the plasma and serum were aliquoted and stored at −80 °C. The frozen samples were transported on dry ice by air to Cambridge, United Kingdom, and stored at −80 °C until analysis. The serum samples were analysed at the Nutritional Biomarker Laboratory at the MRC Epidemiology Unit, University of Cambridge, United Kingdom. The serum folate was measured using ultra-performance liquid chromatography coupled to tandem mass spectrometry (UPLC-MS/MS) [20]. This allowed the highly specific detection of six folate forms: 5-methyltetrahydrofolate (5-methylTHF), tetrahydrofolate (THF), 5-formyltetrahydrofolate (5-formylTHF), free folic acid, 5,10 methenyltetrahydrofolate (5,10 methenylTHF), and an oxidation product of 5-methyltetrahydrofolate (pyrazino-s-triazine derivative (meFox)). Solid phase extraction with phenyl columns was used to isolate the folate forms in the serum samples. The analytes were analysed using reversed UPLC on a column (Waters Acquity UPLC^®^ HSS T3 C8 1.7 μm 2.1 × 100 mm, Wilmslow, UK) at 30 °C before mass spectrometry analysis. The addition of stable isotope labelled internal standards during the extraction, which undergo identical processing, helps to normalise for sample preparation and instrument variability. The concentrations of the analytes were determined by comparing the analyte/internal standard signal to that of the calibration curve. 

As >90% of the results were below the limit of detection (<0.05 nmol/L) for 5,10 methenylTHF and 5-formylTHF, and meFox lacks biological activity, we calculated total serum folate as the sum of three folate forms (5-methylTHF, free folic acid, and THF) and used it in the subsequent analysis. To enable comparability with previous studies that used measurement methods that do not separately identify different folate forms, we also calculated “total serum folate including meFox”, which included the oxidation product meFox. Throughout this article, we use “serum folate” to refer to the total folate calculated without the oxidation product meFox. In total, blood samples from 578 participants were available for folate analysis.

Serum holoTC was measured by a sandwich enzyme-linked immunosorbent assay (ELISA) manufactured by Axis-Shield, Dundee, Scotland. A specific antibody coated on the plate reacted with holoTC. A detection antibody added directly to the plate reacted to form a “sandwich” to produce a colour change. The absorbance was read in a microwave spectrophotometer and the concentration was extrapolated from the calibration curve. The blood samples from 547 participants were available for holoTC biochemical analysis.

### 2.2. Assessment of Fruit and Vegetable Intake

The data on fruit and vegetable intake were collected by trained interviewers using an adapted version of the WHO STEPwise approach to Surveillance (STEPS) questionnaire [21]. The participants were asked four questions relating to the frequency of their fruit and vegetable intake: (i) the number of days in a typical week when they ate fruit or (ii) vegetables; and (iii) the number of times in a typical week they ate fruit or (iv) vegetables.

### 2.3. Measurement of Covariates

The data on socio-demographics (age, sex, education level, rural or urban residence) and health behaviour (alcohol intake, smoking, physical activity) were also collected by the interviewers using an adapted version of the WHO STEPS questionnaire, as for the fruit and vegetable intake [21]. Smoking and alcohol intake were categorised as never, past or current. Blood pressure was measured using an automated blood pressure measuring device (OMRON M4-1) on the participants’ dominant arm after at least 5 min of rest. Three measurements of blood pressure were taken at one-minute intervals and the blood pressure value was computed as the average of the three recordings. Waist circumference was measured to the nearest 0.1 cm, with the participants wearing light clothing, using a non-stretch fiberglass tape at the level of the midpoint between the lower costal margin and the anterior superior iliac crests; heights were measured using a standard rigid stadiometer. Bodyweight and composition were measured using electronic scales and bioelectrical impedance (Tanita TBF-531 scales; Tanita UK, Uxbridge, Middlesex, United Kingdom), respectively. Body mass index (BMI in kg/m^2^) was computed as the body weight (kg) divided by the square of the height (m^2^). 

The physical activity data were collected with self-reported and objective methods. We used the global physical activity questionnaire (GPAQ) and derived estimates in MET-min/week of energy expenditure in different domains (work, leisure, and travel) and overall physical activity energy expenditure (GPAQ PAEE) [21]. Physical activity energy expenditure (PAEE) was measured objectively over seven continuous days using a combined heart rate and movement sensor (Actiheart; Cambridge Neurotechnology, Cambridge, UK). The validity of this method was assessed in this population against PAEE measured with doubly labelled water (*r* = 0.40) and is detailed elsewhere [22]. PAEE scaled for body weight was expressed as KJ/Kg/day after calibration using individual heart rate. Categories were created based on time spent in minutes per day at different intensities of physical activity: <1.5 metabolic equivalents of task (METS), sedentary behaviour; 1.5–3 METS, light physical activity (LPA) >3 METS, moderate-to-vigorous physical activity [23]. Throughout, we use PAEE to refer to objectively measured physical activity. 

### 2.4. Other Biochemical Measurements

Fasting and 2 h glucose post-75 g oral glucose tolerance test were measured on fresh capillary whole blood using a Hemocue B-Glucose analyser (HemoCue AB, Ängelholm, Sweden) onsite. The fasting blood samples were stored and transported to Cambridge in −80 °C storage and the aliquots were used to measure insulin by fluorometric assay on a 1235 AutoDELFIA automatic immunoassay system (kit by Perkin Elmer Life Sciences; Wallac Oy, Turku, Finland). The total cholesterol, HDL, and triglycerides were measured by enzymatic method using automated assays on the Dade Behring Dimension RxL analyser. The LDL concentrations were derived using the Friedewald formula (LDL = total cholesterol − (triglyceride/2.2) − HDL), when triglyceride levels were <4.5 mmol/L. These analyses were conducted at the National Institute for Health Research (NIHR) Cambridge Biomedical Research Centre (BRC), Core Biochemical Assay Laboratory.

### 2.5. Outcome Measurement

A metabolic syndrome z-score was computed by summing the sex-specific standardised values of fasting glycaemia, blood pressure ([(systolic blood pressure + diastolic blood pressure)/2], waist circumference, triglycerides, and HDL cholesterol, with the latter in an opposite direction to the others (i.e., inverted HDL cholesterol) and used in subsequent analysis. Each individual component of the metabolic syndrome was standardised by subtracting the sample mean from individual values and dividing it by the standard deviation of the sample mean. A higher metabolic syndrome score was indicative of a less favourable metabolic profile. We used a continuously distributed variable to avoid the loss of information that would result from categorisation and thus maximise statistical power. In addition, increasing evidence supports the use of a continuous metabolic syndrome score rather than a dichotomous variable [1]. We calculated the homeostatic model assessment of insulin resistance (HOMA-IR) using the formula = ([FPI × FBG]/22.5)), where FPI is fasting plasma insulin (mU/L) and FBG is fasting blood glucose (mmol/L) [24]. We also computed a metabolic syndrome score omitting waist circumference.

### 2.6. Statistical Analysis

All the statistical analyses were performed using Stata 15 (Statacorp, College Station, TX, USA). The data are presented as the mean (SD) for continuous variables (or median (25th–75th percentile) for non-normally distributed variables) and percentages for categorical variables. Using previously suggested cut-offs, we reported the proportion of participants with folate deficiency (serum folate < 10 nmol/L) and vitamin B12 deficiency (holoTC < 50 pmol/L) [25,26]. We tested differences in means using the t-test (or differences in medians using the Mann–Whitney test) and differences in proportions using the chi-squared test. The Spearman coefficient was used to assess pairwise correlations between serum folate and holoTC and between serum folate and self-reported fruit and vegetable intake. We fitted linear regression models adjusted for age and sex to identify predictors of serum folate and holoTC after the log transformation of serum folate to account for its skewed distribution. 

We searched for non-linear relationships between serum folate, holoTC, and outcomes by fitting restricted cubic splines with five knots at the 5th, 27.5th, 50th, 72.5th, and 95th percentiles. As the tests for non-linearity were non-significant, we fitted multiple linear regressions using a block-wise selection approach to estimate the β coefficients and 95% confidence intervals (CI) per one standard deviation (SD) of B-vitamins concentrations. Three models with incremental adjustment for potential confounders were used. Model 1 was unadjusted; model 2 was adjusted for age, sex, level of education (less than primary school, completed primary school, secondary school and university), smoking status (never smoked, past, and current smoker), alcohol intake (never, past, and current) and residential site (four sites); Model 3 was additionally adjusted for BMI (continuous) and PAEE (continuous). Participants with missing data on covariates were excluded from the analysis. We tested the interactions between the B-vitamins and sex, BMI categories and residential site using Model 3 and performed subgroup analyses if the *p*-value for interaction was <0.05.

In the sensitivity analyses, we: (a) fitted linear regression models using serum folate including the oxidation product meFox; (b) mutually adjusted for the other B-vitamins in model 3 and tested for interaction between the B-vitamins; (c) used a metabolic syndrome score computed without waste circumference to adjust for adiposity when waist circumference was not included in the outcome score; (d) used multiple imputation to investigate the impact of missing data on the results. We used multiple imputation by chained equations under the assumption of missing at random, created 10 multiply imputed datasets, and then used Rubin’s combination rules to combine estimates [27]. Throughout, a two-sided α level of 0.05 was used to test for statistical significance. 

## 3. Results

In total, blood samples were available from 578 and 547 participants for the analysis of folate and holoTC, respectively. The demographics, health-related behaviour, and biochemical characteristics of the samples are presented in the Appendix A. The mean age ± SD of the participants was 38.2 ± 8.6 years. The median concentration for serum folate was 12.9 (IQR: 8.6–20.5) nmol/L and the mean of holoTC was 75 ± 34.3 pmol/L. Rural residents demonstrated higher concentrations of serum folate (15.9 (IQR: 9.8–25.9 nmol/L) than those living in urban areas (11.3 (7.9–15.8 nmol/L), *p*-value < 0.0001. The distributions of both B-vitamins were similar in men and women in the entire sample, but serum folate was higher amongst rural men (18.1(11.3–27.1) nmol/L than rural women (14.4(9.3–24.3) nmol/L. Participants living in urban areas demonstrated higher serum holoTC concentrations (79.8 ± 34.9 pmol/L) than rural residents (69.8 ± 32.9 pmol/L), *p*-value= 0.0006. There was a negative correlation between serum folate and holoTC (*r*: −0.12 (*p*-value = 0.007)). In total, 35.3% of the participants were deficient in folate (serum folate <10 nmol/L) and 26.5% deficient in vitamin B12 (holoTC <50 pmol/L). The characteristics of the participants with both serum folate and holoTC deficiency were similar to those of the rest of the sample (Appendix A). 

The median (IQR) number of times people self-reported consuming fruit in a typical week was 2 (1–5) times/week and the comparable figure for vegetables was 4 (2–7) times/week. Rural residents reported higher frequency of fruit consumption (3 (1–6) times/week) than urban residents (2 (1–4) times/week), *p* = 0.005. Similarly, the frequency of self-reported vegetable consumption was higher among rural residents (5 (2–9) times/week than among those living in urban areas 4 (2–6) times/week *p* < 0.0001 (Appendix A). Women reported a higher frequency of consumption of both fruit (3 (1–6) times/week) and vegetables (4 (2–8) times/week) than men, among whom comparable frequencies were 1 (2–4) times/week and 3 (2–6) times/week, respectively (*p* < 0.001 for both comparisons) (Appendix A).

After controlling for age and sex, we observed positive correlations between serum folate and alcohol intake and PAEE, while education level and BMI were inversely associated. For holoTC, positive associations were found for age, education level, and BMI, while PAEE was inversely associated (Table 1). Serum folate was not correlated with self-reported intake of fruit (*r*: 0.003 (*p*-value = 0.95)) and vegetables (r: 0.05 (*p*-value =0.24)).

Serum folate was inversely associated with the metabolic syndrome score in unadjusted analysis (β: −0.30 (95% CI, −0.51 to −0.09) per 10.8 nmol/L (1 SD) of serum folate). This remained significant in Model 3, adjusted for age, sex, education level, smoking, alcohol intake, area of residence, BMI and PAEE (β: −0.20 (95% CI, −0.38 to −0.02) per 1 SD of serum folate) (Table 2, Appendix A)). For individual risk factors, serum folate was inversely associated with diastolic blood pressure (−1.13 (−2.04 to −0.21) per 1 SD of folate) and positively associated with HDL cholesterol (0.04 (0.005 to 0.07) in Model 3.

For serum holoTC, positive unadjusted associations were observed with the metabolic syndrome score (β: 0.33 (95% CI, 0.10 to 0.56) per 34.3 pmol (1 SD) of holoTC), diastolic BP (1.50 (0.39 to 2.62)) and 2 h glucose (0.22 (0.05 to 0.40)) (Table 3, Appendix A)). These associations were attenuated and became non-significant after adjusting for socio-demographic and behavioural characteristics (model 2). There was no evidence of interaction between sex, rural/urban residence, BMI categories, and B-vitamins and any outcome. 

In the sensitivity analyses, (a) the inverse association between serum folate and the metabolic syndrome score was attenuated and became non-significant when we additionally adjusted for serum holoTC in model 3 (β: −0.15 (95% CI, −0.34 to 0.03) per 1 SD of serum folate). There was no evidence of interaction between serum folate and holoTC on the metabolic syndrome score in Model 3 (*p*-value for interaction = 0.39). The inverse association between serum folate and the metabolic syndrome score was similar when we used (b) total folate, including the oxidation product meFox, (c) multiple imputation to investigate the impact of missing data, and (d) a computation of the metabolic syndrome score omitting the waist circumference.

## 4. Discussion

In this cross-sectional, population-based study, serum folate and holoTC were associated with the clustered score of cardiometabolic risk factors in opposite directions. The positive association between serum holoTC and metabolic syndrome score was confounded by socio-demographic and behavioural characteristics. The inverse association between serum folate and metabolic syndrome score was independent of age, sex, education level, smoking, alcohol intake, residential site, BMI, and physical activity. Moreover, higher serum folate was associated with higher HDL and lower diastolic blood pressure independently of potential confounders. As serum folate and holoTC status are dependent on dietary patterns, the observed associations reflect the role of dietary patterns for which folate and vitamin B12 are objective indicators. To our knowledge, this is the first study from SSA to examine the association between the metabolic syndrome and serum folate and holoTC, a marker of vitamin B12 status. Similar to previous studies, we observed a less favourable metabolic profile in participants living in urban areas compared with those living in rural areas [28,29]. This rural-urban difference may partly be attributed to the lower physical activity levels [19] and unhealthy diets [30] reported in urban settings compared with rural settings. 

Previous observational studies in Western and Eastern populations have examined the association of blood concentrations of folate and vitamin B12 with the metabolic syndrome. In a cross-sectional population-based study of 2201 adults in the US, participants with metabolic syndrome demonstrated a higher folate concentration and lower vitamin B12 concentration than those without metabolic syndrome [31]. Similarly, a study of 524 adults in nine Mesoamerican countries showed that red blood cell (RBC) folate was positively associated with the metabolic syndrome and plasma vitamin B12 was positively associated with fasting blood glucose and hypertension [32]. In another study on obese individuals (BMI > 35 kg/m^2^) in France, plasma folate and vitamin B12 showed no association with the number of metabolic syndrome components [33]. Other studies examining the association of dietary intake of folate and B12 with the metabolic syndrome found an inverse association with folate intake [34,35] and no association with vitamin B12 intake [35].

While the evidence from epidemiological studies on the association between these B-vitamins and the metabolic syndrome is conflicting, the difference in the ascertainment of the exposure should be noted. Some studies measured serum (or plasma) folate/vitamin B12, others RBC folate and others dietary folate/B12 intake which may limit comparability across studies. For instance, RBC folate reflects body stores and is thus an indicator of longer-term status (120 days), while serum folate responds rapidly to changes in diet. Moreover, the relationship between folate/B12 intake and status markers is influenced by several factors, such as polymorphisms in metabolising enzymes, smoking, physical activity, vitamin absorption, bioavailability from natural food sources, fortification, and food processing [36]. The positive association between serum folate and the metabolic syndrome reported in the US may not reflect the association of dietary sources of folate with metabolic endpoints given that folic acid fortification of wheat flour has been mandatory in the US since 1998 [37]. 

In this study, we observed an inverse association between serum folate and metabolic syndrome and a positive association with serum holoTC, which reflect the associations of metabolic syndrome with diet. Green leafy vegetables, legumes, and fruits are rich sources of folate and serum folate has therefore been proposed as an objective indicator of fruit and vegetable intake with evidence from observational and intervention studies [12,13,14,15]. Our findings of an inverse association between serum folate and the metabolic syndrome and diastolic blood pressure confirm previously reported cross-sectional studies in SSA showing an inverse association between a dietary pattern rich in fruit and vegetable intake and the metabolic syndrome [38] and hypertension [5,6,7,38].

There are several potential mechanisms through which fruit and vegetable intake may reduce metabolic syndrome risk. Firstly, apart from being rich in folate, fruit and vegetables are also rich sources of vitamin C, E, carotenoids, magnesium and phytochemicals like polyphenols. A higher fruit and vegetables intake has been associated with lower inflammatory markers and reactive oxygen species [39] thereby reducing oxidative stress and systemic inflammation, which are involved in the development and severity of the metabolic syndrome [40]. Secondly, fruits and vegetables contain polyphenols and fibres, which modulate the gut microbiome composition and function [41], and metabolic syndrome has been linked with altered gut microbiota [42],; moreover, dietary fibres offer benefits for individual components of the metabolic syndrome [43]. Thirdly, increasing fruit and vegetable intake is associated with weight loss and adiposity plays a significant pathophysiological role in the development of the metabolic syndrome [44].

While serum folate may be an objective indicator of fruit and vegetable intake, we cannot exclude the possibility that the observed association was due to the metabolic effects of folate via its lowering effect on plasma homocysteine. Folate is required for the conversion of homocysteine to methionine; in case of folate deficiency, the reaction is inhibited, leading to the accumulation of homocysteine. Elevated plasma homocysteine has been linked with a higher prevalence of the metabolic syndrome [45,46]. High plasma homocysteine levels can be lowered by folic acid supplementation [ref] but evidence of the efficacy of folic acid supplementation in lowering cardiometabolic risk is inconclusive. A meta-analysis of randomised controlled trials showed that folic acid supplementation lowered the risk of cardiovascular disease [47]. In another meta-analysis, folic acid supplementation did not lower diabetes risk [48]. In this study, the inverse association between serum folate and the metabolic syndrome score became non-significant after adjusting for holoTC. One reason for this could be because vitamin B12 together with folate act as cofactors in the remethylation of homocysteine to methionine, therefore the mutual adjustment of these B-vitamins in the analyses may attenuate or mask potential associations [49]. 

We found a positive association between holoTC and the metabolic syndrome in our unadjusted analysis, which could suggest a potential positive association of the metabolic syndrome with animal-sourced foods. Natural vitamin B12 is found almost exclusively in animal products, which are high in iron and fats especially saturated fats. High intake of animal-based fat has been linked with higher risk of obesity, hyperglycaemia, and cardiovascular disease. Moreover, iron features pro-oxidative properties and may contribute to increasing oxidative stress through the generation of reactive oxygen species [50]. The positive association between holoTC and the metabolic syndrome became non-significant after adjusting for potential confounders, suggesting that socio-demographic variations and health-related behaviours contributed to the association observed.

Serum folate concentrations were higher in rural residents than in those living in urban areas, while holoTC was higher in urban areas. This is consistent with findings from a meta-analysis of 47 studies from 22 countries in SSA showing a higher meat intake in urban residents compared with those living in rural areas, and though not statistically significant, a tendency towards higher intake of fruit and vegetables in rural areas [8]. Although there was no correlation between serum folate and the self-reported frequency of fruit and vegetable consumption, the socio-demographic patterning of self-reported frequency of fruit and vegetable consumption was similar to that of serum folate. We observed a higher self-reported frequency of fruit and vegetable consumption in rural residents compared with those living in urban areas. The frequency of fruit and vegetable consumption was based on self-reports, in which the participants reported the number of times in a typical week when they ate fruit and vegetables, which could have been influenced by recall bias and social desirability bias. Moreover, portion sizes and detailed information about the fruits and vegetables consumed (for instance, fresh vegetables, dried vegetables, cooked or raw vegetables) were not estimated. We also did not obtain evidence for the folate content of the fruit and vegetables consumed and whether this content is different in rural versus urban settings. All these factors could affect the relationship with serum folate. In a nationally representative survey of women of reproductive age in Cameroon, fruit, vegetables, beans and legumes, grains, fruit, roots, and tubers contributed 98% of the total folate intake [51]. 

Future studies are needed to determine which dietary changes can provide a sustainable change in folate status and whether these changes exert a positive effect on the reduction of metabolic risk.

## 5. Strengths and Limitations

The strengths of our study include the objective assessment of the B-vitamins in a population-based study. Folate status was measured using UPLC-MS/MS, which is regarded as the gold standard for serum folate and captures the individual folate forms separately, including free folic acid. We measured holoTC, which has been postulated as a better marker of vitamin B12 status than total vitamin B12 [18]. Moreover, our study included participants from rural and urban areas and we adjusted for a wide range of potential confounders, including objectively measured physical activity. Physical activity may lead to a higher turnover of B-vitamins because of increased protein catabolism [52]. In our study, physical activity was positively associated with serum folate but inversely associated with serum holoTC, suggesting that the association of B-vitamins with physical activity may partly reflect differences in health-related behaviours rather than a mechanistic association. For example, a higher intake of fruit and vegetable and a lower intake of animal-sourced foods in more physically active individuals. 

Our study featured some limitations. Firstly, this was an observational study and we cannot rule out the possibility of residual confounding or confounding from unmeasured factors. Secondly, the cross-sectional design of the study limited our ability to establish temporality, and the possibility of reverse causation cannot be excluded. Thirdly, the samples used in this study were collected and stored in 2005–2006. However, both folate and vitamin B12 have been shown to be relatively stable after long-term storage [53]. Moreover, because, these samples were collected before the mandatory wheat flour fortification in Cameroon in 2011, our study provides a unique opportunity to examine the associations of serum folate and vitamin B12 as objective indicators of fruits and vegetable intake and animal-sourced foods intake, respectively, with cardiometabolic risk factors. Fourthly, detailed data on fruit and vegetable intake were not available and there was no data on animal product intake. However, previous observational and intervention studies have reported an association between serum folate and intake of fruit and vegetables and between plasma vitamin B12 and intake of animal products. This suggests that folate and B12 may be used as objective indicators of dietary intake, especially in settings where there is no food fortification with these B-vitamins. Lastly, we collected data on apparently healthy adults in Cameroon aged 25–55 years and our findings may not be generalisable to an unhealthy population or a population outside this age range or geographical region. Our results show that it is important to extend this work to other age ranges and at-risk populations to enable a more targeted approach to quell the rapid rise of metabolic syndrome in SSA.

## 6. Conclusions

In this population-based study, higher serum folate as an objective indicator of fruit and vegetable intake was associated with lower composite metabolic syndrome score, diastolic blood pressure, and higher HDL cholesterol, independently of socio-demographic and health-related behaviours. Higher holoTC concentration as an indicator of intake of animal-sourced foods was associated with a higher metabolic syndrome score, but this association was confounded by sociodemographic characteristics. This suggests that, based on these biomarkers, public health approaches promoting a higher intake of fruit and vegetables may reduce the burden of cardiometabolic disease in this population. Future large and prospective studies to investigate the associations of serum folate and vitamin B12 with metabolic risk in SSA populations are needed, as well as studies to explore the dietary determinants of blood folate and vitamin B12 concentrations.

## Figures and Tables

**Table 1 nutrients-14-00178-t001:** Factors associated with serum folate (*n* = 578) and holotranscobalamin (*n* = 547); Cameroon study.

Characteristics	Folate Median (IQR) = 12.9 (8.6–20.5) nmol/L β (95 Confidence Interval)	*p*-Value	Holotranscobalamin Mean ± SD = 75 ± 34.3 *p*mol/L β (95 Confidence Interval)	*p*-Value
Age (years)	0.10 (0.04 to 0.16)	0.001	5.5 (2.1 to 8.9)	0.002
Men (vs women)	0.06(−0.05 to 0.18)	0.273	0.36 (−5.71 to 6.43)	0.908
Education level (completed) <primary education (ref) Primary Secondary and high school University	0.04(−0.11 to 0.19) −0.12(−0.29 to 0.05) −0.37(−0.57 to −0.18)	0.639 0.176 <0.001	4.15(−3.58 to 11.88) 9.9(1.29 to 18.69) 13.83(3.35 to 24.31)	0.292 0.024 0.010
Urban (vs rural)	−0.29(−0.39 to −0.19)	<0.001	10.8(5.1 to 16.4)	<0.001
Smoking status Never smoked (ref) Former smoker Current smoker	0.08(−0.08 to 0.24) 0.02(−0.21 to 0.25)	0.359 0.847	0.78(−8.07 to 9.63) 4.73(−7.34 to 16.79)	0.863 0.442
Alcohol drinking Never (ref) Former Current	0.01(−0.21 to 0.22) 0.19(0.04 to 0.34)	0.931 0.014	−3.82(−14.62 to 6.99) 7.50(−0.99 to 15.99)	0.488 0.083
Fruits (times/week)	0.003 (−0.01 to 0.02)	0.741	0.28(−0.61 to 1.16)	0.54
Vegetables (times/week)	0.001 (−0.01 to 0.01)	0.897	−0.46(−1.21 to 0.29)	0.227
PAEE (KJ/Kg/day)	0.005(0.003 to 0.008)	<0.001	−0.18(−0.32 to −0.04)	0.011
Sedentary (hour/day)	−0.04 (−0.06 to −0.02)	<0.001	0.81 (−0.48 to 2.1)	0.216
LPA (hour/day)	0.03 (0.001 to 0.07)	0.041	0.15 (−1.77 to 2.08)	0.874
MVPA (hour/day)	0.06 (0.03 to 0.10)	0.001	−2.64 (−4.73 to −0.54)	0.014
GPAQ PAEE (KJ/Kg/day)	0.001(0.0002 to 0.001)	0.007	−0.01(−0.05 to 0.02)	0.471
GPAQ work (MET-h/week)	0.0005(0.0001 to 0.0008)	0.013	−0.004(−0.02 to 0.02)	0.702
GPAQ leisure (MET-h/week)	0.0005(−0.002 to 0.003)	0.622	−0.03(−0.11 to 0.05)	0.463
GPAQ travel (MET-h/week)	0.001(−0.0004 to 0.003)	0.141	−0.06(−0.13 to 0.008)	0.083
BMI (Kg/m^2^)	−0.01(−0.02 to −0.004)	0.005	1.21(0.65 to 1.77)	<0.001
BMI categories (Kg/m^2^) <25 (ref) 25–29.9 ≥30	−0.15(−0.26 to −0.03) −0.17(−0.29 to −0.04)	0.015 0.011	5.17(−1.81 to 12.15) 13.22(5.63 to 20.82)	0.146 0.001
Body fat (10%)	−0.09(−0.20 to −0.03)	0.004	10.0(6.8 to 14.1)	<0.001
Waist circumference (10 cm)	−0.06(−0.10 to −0.02)	0.003	4.1 (1.6 to 6.7)	0.002

CI, confidence interval; PAEE, physical activity energy expenditure; LPA, light physical activity; MVPA, moderate-to-vigorous physical activity; GPAQ, global physical activity questionnaire; BMI, body mass index. β-coefficients represent the difference in holotranscobalamin in pmol/L or the log transformed value of folate in nmol/L per a unit difference in the predictor. For example, a β-coefficient of 0.01 for folate means that, for a year difference in age, folate concentration changes by exp (0.01) = 1.01 nmol/L, which corresponds to an increase of 1.0%. Estimates were adjusted for age and sex (except for age adjusted for sex only and sex adjusted for age only).

**Table 2 nutrients-14-00178-t002:** Associations between serum folate and metabolic risk factors; Cameroon study.

Outcome	Difference in Outcome per 1 SD (10.8 nmol/L) of Serum Folate					
	Model 1		Model 2		Model 3	
	β (95% CI)	*p*-Value	β (95% CI)	*p*-Value	β (95% CI)	*p*-Value
Metabolic syndrome score (*n* = 520)	−0.30 (−0.51 to 0.09)	0.005	−0.30 (−0.50 to −0.10)	0.004	−0.20 (−0.38 to −0.02)	0.029
Systolic blood pressure (mmHg), (*n* = 529)	−1.51 (−3.18 to 0.16)	0.076	−1.58 (−3.29 to 0.12)	0.069	−1.25 (−2.89 to 0.39)	0.135
Diastolic blood pressure (mmHg), (*n* = 529)	−1.57 (−2.58 to −0.56)	0.002	−1.42 (−2.39 to −0.45)	0.004	−1.13 (−2.04 to −0.21)	0.016
Fasting blood glucose (mmol/L), (*n* = 529)	−0.003 (−0.10 to 0.10)	0.957	0.01 (−0.08 to 0.11)	0.807	0.03 (−0.08 to 0.12)	0.629
2-h blood glucose (mmol/L), (*n* = 522)	−0.01 (−0.17 to 0.15)	0.865	0.01 (−0.15 to 0.18)	0.881	0.02 (−0.15 to 0.19)	0.809
HOMA_IR, (*n* = 526)	−0.004 (−0.09 to 0.09)	0.932	0.02 (−0.06 to 0.11)	0.603	0.05 (−0.03 to 0.14)	0.224
HDL cholesterol (mmol/L), (*n* = 520)	0.03 (0.002 to 0.06)	0.037	0.04 (0.006 to 0.07)	0.018	0.04 (0.01 to 0.07)	0.023
Triglycerides (mmol/L), (*n* = 520)	0.0004 (−0.04 to 0.04)	0.983	−0.01 (−0.05 to 0.03)	0.525	−0.003 (−0.04 to 0.04)	0.883

HOMA-IR, homeostatic model assessment of insulin resistance; β, β-coefficient; CI, confidence interval. Model 1: Unadjusted. Model 2: Adjusted for age, sex, education level, smoking, alcohol intake, residential site. Model 3:Model 2 + body mass index + objectively measured physical activity.

**Table 3 nutrients-14-00178-t003:** Associations between serum holotranscobalamin (holoTC) and metabolic risk factors; Cameroon study.

Outcome	Difference in Outcome per 1 SD (34.3 pmol/L) of Serum holoTC					
	Model 1		Model 2		Model 3	
	β (95% CI)	*p*-Value	β (95% CI)	*p*-Value	β (95% CI)	*p*-Value
Metabolic syndrome score, (*n* = 491)	0.33 (0.10 to 0.56)	0.004	0.17 (−0.05 to 0.39)	0.127	0.04 (−0.16 to 0.23)	0.70
Systolic blood pressure (mmHg), (*n* = 500)	1.72 (0.001 to 3.44)	0.050	0.49 (−1.16 to 2.14)	0.561	0.08 (−1.60 to 1.76)	0.923
Diastolic blood pressure (mmHg), (*n* = 500)	1.50 (0.39 to 2.62)	0.008	0.65 (−0.42 to 1.71)	0.234	0.32 (−0.75 to 1.38)	0.561
Fasting blood glucose (mmol/L), (*n* = 500)	0.10 (−0.03 to 0.22)	0.121	0.07 (−0.07 to 0.21)	0.323	0.05 (−0.09 to 0.18)	0.502
2-h blood glucose (mmol/L), (*n* = 494)	0.22 (0.05 to 0.40)	0.014	0.12 (−0.05 to 0.30)	0.162	0.12 (−0.06 to 0.29)	0.187
HOMA-IR, (*n* = 497)	0.09 (−0.003 to 0.18)	0.059	0.12 (0.03 to 0.23)	0.012	0.08 (−0.01 to 0.18)	0.075
Triglycerides (mmol/L), (*n* = 491)	0.03 (−0.006 to 0.07)	0.10	0.02 (−0.02 to 0.06)	0.313	0.01 (−0.03 to 0.05)	0.475
HDL cholesterol (mmol/L), (*n* = 491)	0.02 (−0.01 to 0.05)	0.165	0.01 (−0.02 to 0.03)	0.678	0.01 (−0.02 to 0.04)	0.647

HOMA-IR, homeostatic model assessment of insulin resistance; β, β-coefficient; CI, confidence interval. Model 1: Unadjusted. Model 2: Adjusted for age, sex, education level, smoking, alcohol intake, residential site. Model 3: Model 2 + body mass index + objectively measured physical activity.

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
