# Peer review of "Associations of Serum Folate and Holotranscobalamin with Cardiometabolic Risk Factors in Rural and Urban Cameroon"

_nutrients, 2021, doi:10.3390/nu14010178_

Round 1
Reviewer 1 Report
I would like to congratulate the authors of the manuscript " Associations of serum folate and holotranscobalamin with cardiometabolic risk factors in rural and urban Cameroon". I have only one comment/question.
Lines: 365-367: You wrote about the positive association between holoTC and the metabolic syndrome and suggested a potential inverse association of MS with animal-sourced foods. Is it correct? If yes, explain it, please.
Reviewer 2 Report
The article entitled “Associations of serum folate and holotranscobalamin with cardiometabolic risk factors in rural and urban Cameroon” by Mba, et al. It is a cross-Sectional study who has as originality use the Holo-TC as marker of status of B12 and total folates measured by LC-MSMS. They correlated these 2 markers to a Metabolic Syndrome z-score in two population, one rural et one urban.
It is an interest article, but it need some upswings
Line 67-68
“HoloTC is the form of vitamin B12 that can be readily taken up by the cells”
The HoloTC is not a form of vitamin B12. The 2 carriers vitamin B(12)-specific proteins are predominantly transcobalamin (TC) and haptocorrin (HC), of them one biologically inactive haptocorrin-vitamin B12 complex which represents 80 to 90% of the total amount of vitamin B12 and, and the other one, the transcobalamin-vitamin B12, a biologically active complex called holotranscobalamin (Holo-Tc), which constitutes 10-20% of total vitamin B12.
Please replace "form" as appropriate ("biologically active complex (B12-transcobalamin)" or other, so that non-specialist readers would understand the metabolism of B12.
Lines 126-131
“2.2. Assessment of fruit and vegetable intake
Data on fruit and vegetables intake were collected by trained interviewers using an adapted version of the WHO STEPwise approach to Surveillance (STEPS) questionnaire”
Please put the questionnaire in supplementary data
Lines 133-135
“Data on socio-demographics (age, sex, education level, rural or urban residence) and health behaviours (alcohol intake, smoking, physical activity) were also collected by interviewers using an adapted version of the WHO STEPS questionnaire as for fruit and vegetables intake”
Please put the questionnaire in supplementary data
Lines 148-151
“Physical activity data were collected with self-reported and objective methods. We used the global physical activity questionnaire (GPAQ) and derived estimates in METmin/week of energy expenditure in different domains (work, leisure and travel) and over all physical activity energy expenditure (GPAQ PAEE) (21)”
Please put the questionnaire in supplementary data
Lines 174-177
“A metabolic syndrome z-score was computed by summing sex-specific standardised values of fasting glycaemia, blood pressure ([(systolic blood pressure + diastolic blood pressure)/2], waist circumference, triglycerides and HDL cholesterol, with the latter in an opposite direction to the others (i.e inverted HDL cholesterol) and used in subsequent analysis.”
Please define better the z-score for metabolic syndrome, I mean, what is the scale between the best z-score or worst score
Results section
Lines 220-250
Throughout the results, the authors present the correlations in the form of tables or in the body of the text. It would be desirable for the authors to highlight the most significant results in the form of graphs, while keeping their tables.
Line 251
Please put the p-values in the table 1
Lines 258-288
It is possible to represent de metabolic score and the associations with Folates and Holo-TC for the three models in a graphic?
The frequencies of consumption of fruits and vegetables or meat foods are different according to the habitat (rural or urban) or the sex of the participants. In order to enhance the results, the authors should represent them in graphs
Discussion section
The authors should discuss more about the prevalence of metabolic syndrome in their study population. Are there differences between the data published in the literature on the African population?
Reviewer 3 Report
A manuscript by Mba et al. presents results of an interesting study on a generally understudied population. I have only minor suggestions:
- I suggest revising the results for improved clarity.'
- The study is based on blood collected 15 years ago! Please clearly state it as a limitation.
- Please provide the number/date of ethical committee aproval.
